# Natural Thiols, but Not Thioethers, Attenuate Patulin-Induced Endoplasmic Reticulum Stress in HepG2 Cells

**DOI:** 10.3390/toxins13100727

**Published:** 2021-10-14

**Authors:** Hye Mi Kim, Hwa Young Choi, Gun Hee Cho, Ju Hee Im, Eun Young Hong, Hyang Sook Chun

**Affiliations:** 1School of Food Science and Technology, Chung-Ang University, Anseong 17546, Korea; sssw0145@naver.com (H.M.K.); hwayoung0830@gmail.com (H.Y.C.); joo2413@naver.com (G.H.C.); moka5330@naver.com (J.H.I.); 2HK Inno. N Corporation, Eulji-ro, Jung-gu, Seoul 04551, Korea; eyhongs@gmail.com

**Keywords:** patulin, mycotoxin, thiols, ER stress, HepG2 cells

## Abstract

Patulin, a mycotoxin, is known to have cytotoxic effects, but few studies have focused on the involvement of the endoplasmic reticulum (ER) stress response in patulin toxicity and the natural compounds that attenuate it in HepG2 cells. This study tested the ability of patulin to induce ER stress, and that of four thiols and three thioethers to attenuate patulin-induced ER stress in HepG2 cells. Patulin dose-dependently inhibited cell proliferation (IC_50_, 8.43 μM). Additionally, patulin was found to increase the expression levels of ER stress-related genes and/or protein markers, including BiP, CHOP, and spliced XBP1, in HepG2 cells compared to the vehicle control, indicating its potential in ER stress induction. Patulin-induced cytotoxicity in HepG2 cells was reduced by naturally occurring thiol compounds (glutathione, L-acetyl-L-cysteine, cysteine, and captopril), but not by thioether compounds (sulforaphane, sulforaphene, and S-allyl-L-cysteine). Patulin-thiol co-treatment decreased CHOP expression and BiP and CHOP levels in HepG2 cells but did not alter BiP expression. Spliced XBP1 expression was decreased by patulin-thiol co-treatment. Thus, patulin induced ER stress in HepG2 cells and thiols, but not in thioethers, attenuated patulin-induced ER stress.

## 1. Introduction

Patulin (4-hydroxy-4H-furo(3,2-c)pyran-2(6H)-one) is a heat-stable polyketide lactone mycotoxin (Figure 1) [1]. Although a wide variety of fungal species, including *Penicillium*, *Aspergillus*, and *Byssochlamys* species, produce patulin, it is primarily produced by *Penicillium expansum* growth on apples and apple products [1,2].

The contamination of apple products by patulin poses a serious health risk to all consumers, but particularly to children. A US Department of Agriculture survey showed that children consume more apple products during their first year of life (6.4 g/kg body weight/day) than adults do in a year (1 g/kg body weight/day) [3]. This places young children at an increased risk of patulin toxicity. Within the food industry, concerns over patulin contamination are the highest for apples and apple products [4]. Although patulin or patulin-producer contamination has been reported in a variety of other food sources and products, the contamination frequency of these items is much lower than in apples and apple products. An analytical study showed that 18% of juice and 28% of apple-based baby food samples examined contained patulin concentrations beyond the tolerable limits established by the European Union (0.4 μg/kg body weight/day) [4].

Patulin has several toxic effects on animals. The consumption of a quantity ranging from 2.5–41 mg/kg body weight causes severe damage to several organ systems, including the kidneys, intestines, and immune system [5]. The acute symptoms of patulin consumption include anxiety, spasms, dyspnea, pulmonary congestion, edema, ulceration, hyperemia, gastrointestinal tract distension, intestinal bleeding, epithelial cell deterioration, intestinal inflammation, vomiting, and other forms of gastrointestinal and kidney damage [5,6,7,8,9]. The chronic health risks of patulin consumption include neurotoxic, immunotoxic, immunosuppressive, genotoxic, teratogenic, and carcinogenic effects [10,11,12,13,14,15,16]. Evidence of patulin carcinogenicity is considered limited to experimental animals, and patulin is classified as Group 3, or “not carcinogenic to humans”, by the International Agency for Research on Cancer [17].

The toxic effects of patulin on various cells are strongly associated with its effects on the thiol groups of essential biomolecules [9,18]. Patulin is electrophilic and exerts its deleterious effects by covalently binding to cellular sulfhydryl (thiol) groups in proteins and glutathione (GSH). This affinity to thiol groups can reduce the activity of the patulin toxin [19]. Therefore, sulfur-containing compounds are expected to reduce patulin toxicity.

Oxidative stress is defined as a disturbance in the prooxidant-antioxidant balance that may lead to cell damage [20] and is caused by low levels of superoxide anions, hydroxyl radicals, hydrogen peroxide, and unstable intermediates of lipid peroxidation [21]. Several studies have implicated oxidative stress as a mechanism underlying patulin-mediated toxicity. The oxidative damage induced by patulin is deleterious to cells, and exposure to this mycotoxin may pose a risk to humans [22]. Patulin exerts its cytotoxicity through the generation of reactive oxygen species and the activation of endoplasmic reticulum (ER) stress, which can lead to apoptosis.

Organelles that mediate the cell stress response, such as ER, are linked to several human diseases, including diabetes, Parkinson’s disease, and amyotrophic lateral sclerosis [23,24]. ER functions can be disturbed by various insults, such as the accumulation of unfolded proteins and changes in calcium homeostasis [25]. The disruption of ER homeostasis induces ER stress, which leads to the unfolded protein response (UPR). The UPR can be initiated by three ER-resident transmembrane proteins, PRKR-like ER kinase (PERK), inositol-requiring kinase 1 (IRE1), and activating transcription factor 6 (ATF6), which activate distinct signaling cascades that mediate the ER stress response. These three limbs of the UPR activate UPR target genes and proteins to reestablish ER homeostasis [26]. However, prolonged or severe ER stress triggers the mitochondrial apoptotic process to eliminate damaged cells [27].

The functions of the liver include metabolism, detoxification, nutrient storage, and plasma protein and cholesterol synthesis to maintain body homeostasis [28]. However, the involvement of ER stress in the toxic effects of patulin has been reported in colon (HCT116) and kidney cells (HEK293) [22], but rarely in liver cells. Disruption to protein synthesis in this critical organ is fatal. This study investigated whether ER stress is induced in HepG2 cells, an in vitro liver toxicity model, in response to patulin, and, if so, whether it can be alleviated by seven naturally occurring sulfur-containing compounds: four thiols (R-SH), namely GSH, cysteine (Cys), captopril (CAP), and N-acetyl-L-cysteine (NAC); and three thioethers (R-S-R′), namely S-allyl-L-cysteine (SAC), sulforaphane A (SulA), and sulforaphane E (SulE) (Figure 1).

## 2. Results

### 2.1. Effect of Patulin on Metabolic Activity of HepG2 Cells

The cytotoxic effect of patulin on HepG2 cells after 48 h of incubation, as measured using the MTT assay, is shown in Figure 2. Patulin treatment, with concentrations ranging from 1 to 100 µM, markedly decreased cell metabolic activity in a dose-dependent manner. When the cells were treated with high concentrations of patulin (over 100 µM), their metabolic activities did not decrease in a dose-dependent manner; rather, the cells were viable at low levels. The concentration causing a 50% inhibition (IC_50_) was estimated to be 8.43 µM.

### 2.2. Effects of Patulin on the Expression Levels of ER Stress Marker Genes and Proteins

Changes in the mRNA and protein levels of binding immunoglobulin protein (BiP) and cytosine-cytosine-adenosine-adenosine-thymidine (CCAAT)/enhancer-binding protein homologous protein (CHOP), key markers of ER stress, in HepG2 cells were determined using RT-qPCR (Figure 3A,B) and western blot analysis (Figure 3C,D), respectively. The mRNA level of the targeted markers (*BiP* and *CHOP*) was affected by the patulin concentration (10, 25, and 50 µM). As shown in Figure 3A–D, 1 µg/mL of tunicamycin (positive control) induced the significant upregulation of BiP and CHOP mRNA and protein expression in HepG2 cells, which indicated ER stress. Treating HepG2 cells with patulin increased CHOP mRNA and protein levels in a dose-dependent manner (Figure 3B,D). Patulin also increased the BiP mRNA and protein levels in HepG2 cells compared to the vehicle control at concentrations of 10 and 25 µM. However, at a 50 µM patulin concentration, BiP mRNA and protein levels did not increase compared to those in cells treated with vehicle control (Figure 3A,C).

Using RT-qPCR, we studied the effects of patulin on the expression of spliced and unspliced X-box binding protein 1 (XBP1), another key marker of ER stress. To determine whether patulin induced the generation of spliced XBP1 in HepG2 cells, we treated the unspliced XBP1 with the PstI restriction enzyme. The unspliced XBP1 amplified by RT-qPCR was cut by the restriction enzyme PstI and the fragments were assessed. The expression of the spliced XBP1 increased, whereas the levels of the XBP1 digested by PstI decreased at a higher dose of patulin (Figure 3E,F). Spliced XBP1 expression was increased by patulin in a dose-dependent manner. Spliced XBP1 (uncut by PstI) was not observed after treatment with 10 µM patulin (Figure 3F). Taken together, these data suggested that patulin had the potential to induce ER stress in HepG2 cells.

### 2.3. Attenuation of Patulin-Induced Cytotoxicity by Sulfur-Containing Compounds

We investigated the effects of seven sulfur-containing compounds (GSH, Cys, CAP, NAC, SAC, SulA, and SulE) on patulin-induced cytotoxicity in HepG2 cells using an MTT assay. The co-treatment of the HepG2 cells with sulfur-containing compounds (particularly, 62.5 µM or more of GSH, NAC, Cys, and CAP, which contain a thiol group in their structures) and 25 µM patulin increased cell metabolic activity in a dose-dependent manner compared to the control (25 µM patulin alone; Figure 4A). These results suggested that thiols have an inhibitory effect on patulin-induced cytotoxicity. By contrast, the metabolic activity of HepG2 cells co-treated with patulin in the presence of thioethers (SulA, SulE, and SAC) was similar to or lower than that of cells treated with 25 µM patulin alone. SulA and SulE treatment decreased cell metabolic activity to a greater extent than treatment with patulin alone. SAC had no effects on patulin-induced cytotoxicity in the HepG2 cells (Figure 4B).

### 2.4. Effects of Thiol-Containing Compounds on Patulin-Induced ER Stress

The investigation of the effects of four thiols (GSH, Cys, NAC, and CAP) against patulin-induced ER stress at both the mRNA and protein level using RT-qPCR and western blot analysis, respectively, showed that the expression of ER stress-related biomarkers (BiP, CHOP, and spliced XBP1) was increased by patulin treatment. The co-treatment with four thiols at 50 µM attenuated the expression levels of the ER stress-related biomarkers that were increased by patulin exposure.

The co-treatment of the cells with four thiols and patulin changed the expression of BiP, CHOP, and spliced XBP1 compared to their expression after treatment with patulin alone. The BiP mRNA expression decreased slightly when the cells were co-treated with the four thiol-containing compounds (Figure 5A) and patulin. The BiP levels were significantly decreased by co-treatment with patulin and GSH, Cys, or CAP (Figure 5C). The CHOP mRNA and protein levels were decreased by treatment with GSH, Cys, NAC, and CAP (Figure 5B,D). However, NAC and patulin co-treatment did not significantly decrease the protein or mRNA level of BiP. The levels of spliced XBP1 decreased significantly following the co-treatment of the cells with patulin and GSH, Cys, NAC, and CAP (Figure 5E,F). In addition, increased levels of undigested XBP1 with PstI were observed after co-treatment with patulin and GSH, Cys, NAC, and CAP. From these findings, we concluded that thiol-containing compounds decreased or partially inhibited patulin-induced ER stress in HepG2 cells.

## 3. Discussion

This study investigated whether patulin induces ER stress and whether sulfur-containing compounds inhibit patulin-induced ER stress in HepG2 cells. Patulin inhibited the metabolic activity of HepG2 cells in a dose-dependent manner and induced ER stress. In addition, among the seven tested sulfur-containing compounds, thiols (GSH, NAC, Cys, and CAP) markedly reduced patulin-induced cytotoxicity, whereas thioethers (SulA, SulE, and SAC) had no effect. This reduction by thiols may have been related to the decrease in the mRNA and protein expression levels of the molecules associated with ER stress caused by the patulin treatment.

Patulin decreased the metabolic activity of HepG2 cells in a dose-dependent manner (IC_50_ of 8.43 µM) after 48 h of treatment. Other studies on the cytotoxic effects of patulin on HepG2 cells after 24 h incubation indicated the IC_50_ of 15 µM and 9.32 µM [29,30]. In this study, cell viability was measured using an MTT assay, which measures the metabolic activity of enzymatic reduction of tetrazolium compounds to water-insoluble formazans by dehydrogenases occurring in the mitochondria of living HepG2 cells. Although this assay is still used by many researchers to determine cell viability and proliferation, it is reported to have limitations, including interactions with other reducing chemicals, glucose concentrations in the cell culture medium, and involvement with enzymes located in other organelles, such as ER [31]. Considering the longer treatment period, our results roughly followed those of previous reports [29,30]. The in vitro liver cell model used in this study was composed of HepG2 cells. HepG2 cells are non-tumorigenic cells with a high proliferation rate and an epithelial-like morphology that perform many differentiated liver functions and are most commonly used for liver toxicity studies [32]. However, a major drawback is their limited expression of xenobiotic metabolizing enzymes and transporters compared to primary human hepatocytes [32]. Therefore, it is possible that the cytotoxicity reported in this study was somewhat underestimated.

Patulin treatment increased the mRNA and protein levels of ER stress markers (BiP, CHOP, and XBP1) in HepG2 cells. BiP was the first chaperone to be discovered on the ER membrane [33]. BiP modifies unfolded proteins and acts as an ER stress sensor [34]. If a misfolded protein enters the ER, BiP acts as a chaperone by binding directly to the misfolded protein [35] and simultaneously separating from the ER. This process is the initiation of UPR signaling because of ER stress induction and ultimately attenuates ER stress [36]. Excessively high levels of BiP expression delay or prevent the UPR. However, a low level of BiP expression results in premature or prolonged UPR [37]. According to our data, BiP expression was altered in a dose-dependent manner following treatment with 10 and 25 µM of patulin, suggesting that the UPR system is activated by patulin treatment. CHOP is one of the primary effectors of ER stress-induced cell apoptosis [38]. To overcome ER stress, the UPR system is activated by three transmembrane proteins in the ER (IRE1, PERK, and ATF6) and CHOP expression is increased. Increased levels of the CHOP protein activate proapoptotic reactions in the mitochondria and cause cytochrome C release. CHOP overexpression induces cell cycle arrest and apoptosis [39]. Patulin increases CHOP expression and can inhibit HepG2 cell proliferation (Figure 2 and Figure 3). XBP1 expression is induced by ATF6 and XBP1 mRNA and spliced by IRE1 in response to ER stress. Yoshida et al. [40] indicated that the level of spliced XBP1 mRNA resulting from the ER stress induced by 300 nM of thapsigargin is a good marker of ER stress. Unspliced XBP1 has a region targeted by PstI and XBP1 mRNA that is not cleaved by IRE1 activation and can be digested by PstI [41]. Using this reaction, we can more clearly observe whether XBP1 is spliced. A decrease in the levels of XBP1 digested by PstI suggests an increase in the levels of spliced XBP1 following IRE1 pathway activation. The UPR can be activated by patulin at concentrations above 25 µM through the IRE1 pathway. Our results suggested that patulin induces ER stress in HepG2 cells by upregulating the expression of markers associated with ER stress including BiP, CHOP, and spliced XBP1. The expression of spliced XBP1 fragments was induced by patulin (25 and 50 µM), suggesting that patulin triggers ER stress by activating the IRE1 pathway.

The toxic effects of patulin on various cells are strongly associated with its effects on the thiol groups of essential biomolecules [9,18,19]. Thus, we hypothesized that additional treatment with naturally occurring sulfur-containing compounds (GSH, CAP, Cys, NAS, SAC, SulA, and SulE) could reduce patulin-induced toxicity in a cellular model. To examine whether sulfur-containing compounds alleviated patulin-induced cytotoxicity and/or patulin-induced ER stress, we conducted MTT assays and analyzed the mRNA and protein expression levels of BiP, CHOP, and spliced XBP1 using RT-qPCR and western blotting, respectively. Among the sulfur-containing compounds tested, thiols (GSH, NAC, Cys, and CAP) markedly reduced patulin-induced cytotoxicity. However, thioethers (SulA, SulE, and SAC) did not affect the cytotoxicity (Figure 4B). We had assumed that thioethers would also have inhibitory effects on patulin cytotoxicity. Moreover, SulA and SulE thioethers, which differ in their structures only by a single double bond, showed a similar tendency against patulin-induced cytotoxicity, suggesting this relationship is based on structural activity. This study showed that patulin had a strong affinity only for the thiol group.

In terms of mRNA or protein expression, the BiP and CHOP protein levels were decreased by co-treatment with patulin and GSH, Cys, or CAP. The spliced XBP1 was also significantly decreased by co-treatment with all four thiols and patulin. Thiol-containing compounds attenuated patulin-induced ER stress in HepG2 cells. Patulin has an electrophilic property; thus, the nucleophilic part of the thiol group can be attacked by patulin. Patulin reacts with Cys, NAC, and GSH to form patulin-Cys, NAC, and GSH adducts [42]. Morgavi et al. [19] reported that the adducts of patulin and thiol-containing compounds (GSH and Cys) generally have a lower toxicity than patulin alone. Our data suggested that the reduced expression of genes associated with ER stress may have been due to the strong affinity between the thiol group and patulin. From a different viewpoint, patulin is related to oxidative stress. Patulin induces GSH depletion in HepG2 cells. NAC, a GSH synthesis precursor, has an inhibitory effect on patulin-induced chromosome damage because it induces GSH synthesis [43]. Patulin-induced ROS-mediated ER stress in HCT116 and HEK293 human colon carcinoma cells has been reported [22]. Of the thiol-containing compounds, GSH is a strong antioxidant, NAC and Cys are precursors of GSH synthesis [44], and CAP has been reported to exert an antioxidant effect by increasing the GSH levels in a rat model of L-arginine-induced acute pancreatitis [45]. All the thiol-containing compounds chosen for the current study can influence a GSH-associated antioxidant system. Further research is needed to determine whether patulin induces ROS-mediated ER stress and whether compounds with thiol groups attenuate ER stress by modulating GSH levels in HepG2 cells.

To the best of our knowledge, this is the only study to investigate the effects of patulin-induced ER stress in liver cells. The patulin-induced cytotoxicity in the HepG2 cells was dose-dependent, which suggested that patulin can cause ER stress in human hepatocytes. Cytotoxicity and ER stress were decreased by the thiol-containing compounds. We propose that patulin may cause ER stress and promote the death of HepG2 cells via UPR activation. Our findings provide evidence that patulin has potentially injurious effects on human-derived liver cell lines and that thiol-containing compounds (GSH, NAC, Cys, and CAP), but not thioethers, can act as modulators of patulin-induced ER stress. However, because the results of this study were obtained using an in vitro cell model, these effects should further be confirmed in other in vitro hepatic cell models, in vivo animal models or human studies. In addition, the possible roles of the three branches (PERK, ATF6, and IRE1α pathways) in patulin-induced ER stress and the precise mechanism of thiol compounds should be elucidated.

## 4. Materials and Methods

### 4.1. Chemicals and Reagents

The patulin (#P1639), tunicamycin (#T7765), L-GSH (#G4251), NAC (#A9165), CAP (#C4042), L-Cysteine hydrochloride monohydrate (#C6852), 3-(4,5-dimethyl-2-thiazolyl)-2,5-diphenyl-2H-tetrazolium bromide (MTT) (#M2128), and dimethyl sulfoxide (DMSO) (#D8418) were obtained from Sigma-Aldrich (St. Louis, MO, USA). The SAC (#A1468) was obtained from Tokyo Chemical Industry (Tokyo, Japan). The DL-SulA (#sc-207495), L-SulE (#sc-202690), primary antibodies (against phospho-PERK (p-PERK) (#sc-32577), β-actin (#sc-47778)), and secondary antibodies (anti-rabbit (#sc-2004), anti-mouse (#sc-2005)) were obtained from Santa Cruz Biotechnology (Dallas, TX, USA). The primary antibodies against phospho-eIF2α (#119A11), CHOP (#L63F7) and BiP (#C50B12) were obtained from Cell Signaling Technology (Beverly, MA, USA), and the primary antibody against ATF6α (#NBP1-40256) was obtained from Novus Biological (Minneapolis, MN, USA). The electrochemiluminescence (ECL) solution and buffers (such as gel-forming buffer) (#1705061) were obtained from Bio-Rad (Hercules, CA, USA) and used in the western blotting analysis. The polyvinylidene difluoride (PVDF) (#10600023) membranes were obtained from GE Healthcare (Amersham, UK), the radioimmunoprecipitation assay (RIPA) lysis buffer (#MB-030-0050) was obtained from Rockland (Limerick, Ireland), and the Thunderbird SYBR qPCR Mix (#QPS-201) was obtained from Toyobo (Tokyo, Japan). Dulbecco’s modified Eagle’s medium (#11965-092) and related materials were obtained from Gibco Biotechnology (Rockville, MD, USA).

### 4.2. Cell Culture

The HepG2 cell line (ATCC HB-0865™) was obtained from America Type Culture Collection (Manassas, VA, USA). The cells were cultured in DMEM containing 10% heat-inactivated fetal bovine serum and 1% 100 U/mL penicillin-100 µg/mL streptomycin. The cells were incubated in 5% CO_2_ at 37 °C.

### 4.3. Cell Metabolic Activity

The HepG2 cells (2 × 10^4^ cells/mL) were seeded in 96-well plates and incubated for 24 h. After incubation, patulin (1, 5, 10, 25, 50, 100, 150, or 200 µM) or GSH, NAC, Cys, CAP, SAC, SulA, or SulE (each at 3.9, 7.8, 15.6, 31.3, 62.5, 125, or 250 µM) with 25 µM patulin dissolved in 0.5% DMSO were added to the cells in a serum-free medium. Next, the cells were incubated for 24 h or 48 h at 37 °C. Four hours before the end of the treatment, 20 µL of MTT solution in phosphate-buffered saline (5 mg/mL) was added to each well, and the cells were incubated again for the remaining 4 h. The supernatant was removed using suction, and the residue formed by the formazan crystals was dissolved in DMSO. The absorbance was measured at 595 nm using a Thermomax microplate reader (Molecular Devices, San Jose, CA, USA). The data were expressed relative to the vehicle control (0.5% DMSO). The control was normalized as 100% and the cell metabolic activity was analyzed using Origin Pro 8 software (OriginLab Corp., Northampton, MA, USA). The IC_50_ value was calculated from a log concentration of patulin versus cell metabolic activity curve using GraphPad Prism software (version 8.02, San Diego, CA, USA).

### 4.4. Reverse Transcription-Quantitative Real-Time Polymerase Chain Reaction (RT-qPCR) Analysis

The HepG2 cells (60 × 10^4^ cells/well) were seeded in 60-mm culture plates for 24 h and incubated in 5% CO_2_ conditions at 37 °C. The cells were treated with patulin (10, 25, or 50 µM) or GSH, NAC, Cys, CAP, SAC, SulA, or SulE (50 µM) with 25 µM patulin for 12 h. The RNA was extracted from the harvested cells using an RNeasy kit (Qiagen, Hilden, Germany), according to the manufacturer’s instructions. The RNA (1 µg) was reverse-transcribed to cDNA using a QuantiTect Reverse Transcription kit (Qiagen). The total PCR volume (20 µL) contained 50 ng of cDNA, Thunderbird SYBR qPCR Mix, and primer for the target gene. The primer sequences are shown in Appendix A. The experiments were performed using a CFX96 real-time system (Bio-Rad). The PCR conditions were: 40 cycles of pre-denaturation at 95 °C for 3 min, denaturation at 95 °C for 15 s, annealing at 60 °C for 10 s, and extension at 72 °C for 30 s. The mRNA level of the ER stress markers was analyzed using Bio-Rad CFX Manager software (Bio-Rad) and ACTB was the housekeeping reference gene for normalization. The data were expressed relative to vehicle control (0.5% DMSO) expression.

### 4.5. XBP1 Splicing

Amplified XBP1 mRNA and PstI restriction enzyme were added to the reaction buffer and water and the mixture was incubated at 37 °C for 2 h. The PstI restriction site is on an intron of XBP1 mRNA. The mixture was heated at 80 °C for 20 min to inactivate the PstI. The reaction products were loaded onto 4% agarose gels (Dynebio, Seongnam, Korea) with STAR loading solution and electrophoresed at 100 V for 1 h. The DNA fragment pattern was analyzed using a Gel Doc EZ imager (Bio-Rad).

### 4.6. Western Blot Analysis

The HepG2 cells (1.5 × 10^6^) were seeded in a 10 cm culture plate for 24 h. The incubated cells were then treated with 10, 25, or 50 µM patulin for 12 h. The cells were harvested in a RIPA buffer containing 1% protease inhibitor cocktail (Quartett, Berlin, Germany) to destroy the cell membranes and inhibit protein degradation. The lysate was centrifuged at 12,000 rpm (15,520× *g*) for 20 min to extract the protein. The protein samples (40 µg) were loaded into 7.5% or 15% polyacrylamide gel, the individual proteins in the protein mixture were electrophoretically separated, and then the separated proteins were transferred to a PVDF membrane using an SE22 transfer tank (Hoefer, Holliston, MA, USA). The membranes were incubated with primary antibodies targeting BiP (1:1000), cytosine-cytosine-adenosine-adenosine-thymidine (CCAAT)/enhancer-binding protein homologous protein (CHOP, also known as GADD153) (1:1000), and β-actin (1:1000), for 24 h. After washing with TTBS (Tris Buffered Saline, with Tween 20, pH 8.0), the membranes were incubated with secondary antibodies (HRP-conjugated goat anti-mouse and rabbit) for 1 h at 4 °C. The protein expression was measured using a Smart Chemi 500 (Sage Creation, Beijing, China) imager and quantified using Lane ID software (version 4.0).

### 4.7. Statistical Analysis

The data are expressed as the mean ± standard error of the mean (SEM). The statistical analysis was performed using SPSS statistical software (Armonk, NY, USA). The data were analyzed using Student’s *t*-test. The differences between *p*-values < 0.05 were considered statistically significant.

## Figures and Tables

**Figure 1 toxins-13-00727-f001:**
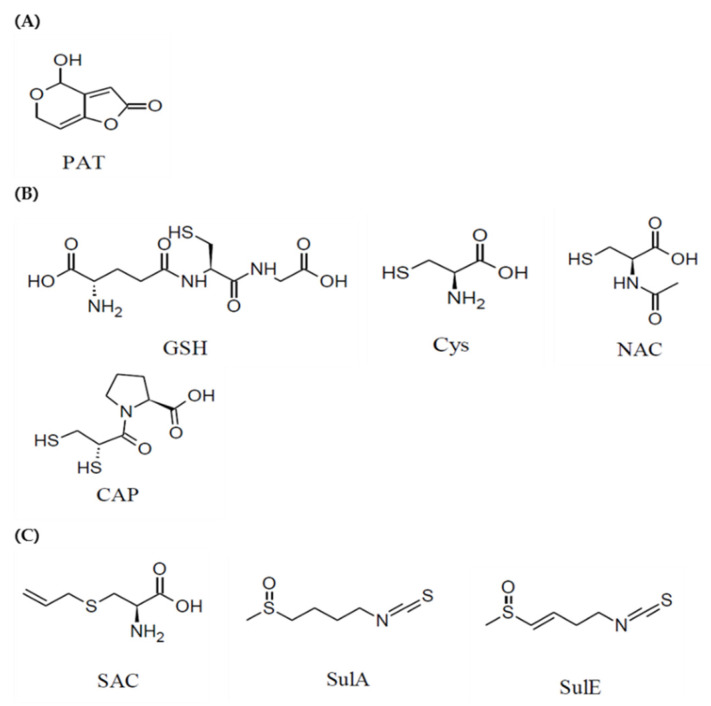
Structure of (**A**) patulin (PAT) and sulfur-containing compounds. The (**B**) thiols tested included glutathione (GSH), cysteine (Cys), N-acetyl-L-cysteine (NAC), and captopril (CAP). The (**C**) thioethers tested included S-allyl-L-cysteine (SAC), sulforaphane A (SulA), and sulforaphane E (SulE).

**Figure 2 toxins-13-00727-f002:**
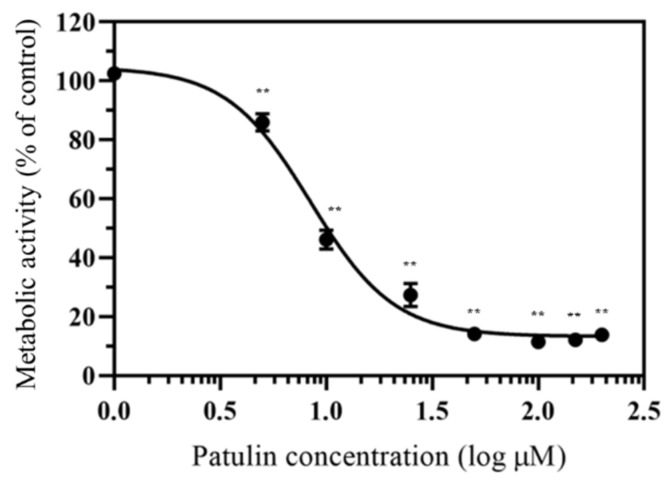
Effect of patulin on HepG2 metabolic activity. Cells were treated with patulin for 48 h. Cell metabolic activity was determined via a 3-(4,5-dimethyl-2-thiazolyl)-2,5-diphenyl-2H-tetrazolium (MTT) assay. IC_50_ was calculated as 8.43 μM. Data are shown as the mean ± SEM from three independent experiments ** *p* < 0.01 (*t*-test) compared with the vesicle control group.

**Figure 3 toxins-13-00727-f003:**
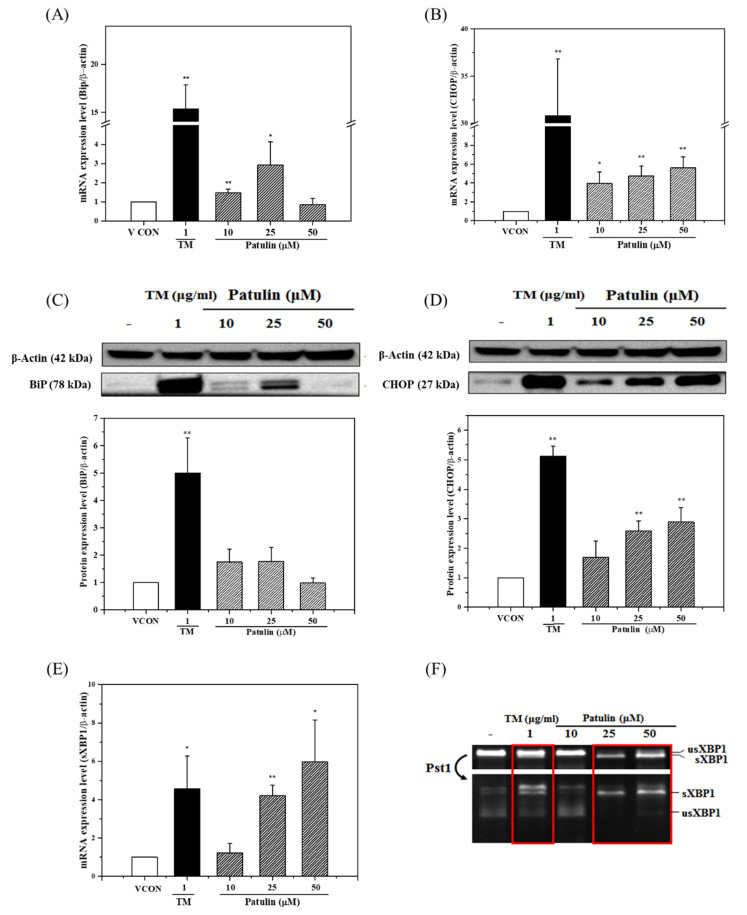
Expression of BiP, CHOP, and XBP1 and their protein levels in HepG2 cells after treatment with patulin (10, 25, and 50 μM) for 48 h. The expression levels of BiP (**A**) and CHOP (**B**) after treatment with patulin were analyzed by reverse transcription-quantitative real-time polymerase chain reaction. The protein expression of BiP (**C**) and CHOP (**D**) after treatment with patulin was analyzed by western blotting. (**E**) The effects of patulin on the expression of spliced XBP1 cut by PstI (**F**) in HepG2 cells usXBP1, unspliced XBP1; sXBP1, spliced XBP1. Tunicamycin (TM) was a positive control. The mRNA for β-actin (*ACTB*) was used as the housekeeping reference gene to normalize the data using the ΔΔCt method. The data are shown as the mean ± SEM from three independent experiments and were compared with the vehicle control (VCON, 0.5% DMSO). * *p* < 0.05 and ** *p* < 0.01 (*t*-test) compared with the vehicle control group.

**Figure 4 toxins-13-00727-f004:**
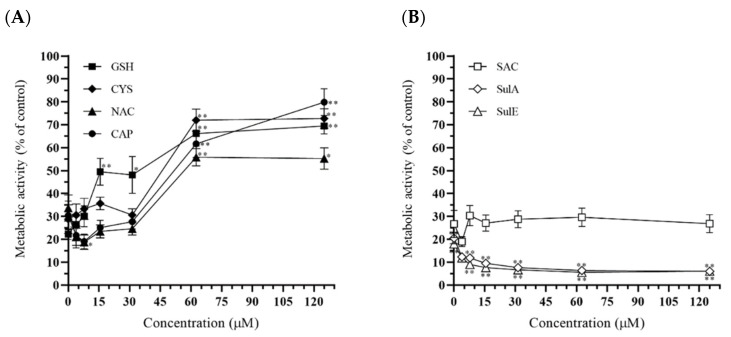
The effect of thiol (**A**) and thioether (**B**) compounds on the metabolic activity of HepG2 cells. The cells were treated with thiols and thioethers s for 48 h. A 3-(4,5-dimethyl-2-thiazolyl)-2,5-diphenyl-2H-tetrazolium (MTT) cytotoxicity assay was performed to determine the metabolic activity of the treated HepG2 cells. GSH, glutathione; NAC, l-acetyl-l-cysteine; Cys, cysteine; CAP, captopril; SAC, sulforaphane; SulA, sulforaphene; SulE, *S*-allyl-L-cysteine. The data are shown as the mean ± SEM from three independent experiments * *p* < 0.05 and ** *p*< 0.01 (*t*-test) compared with the control group (25 μM patulin alone).

**Figure 5 toxins-13-00727-f005:**
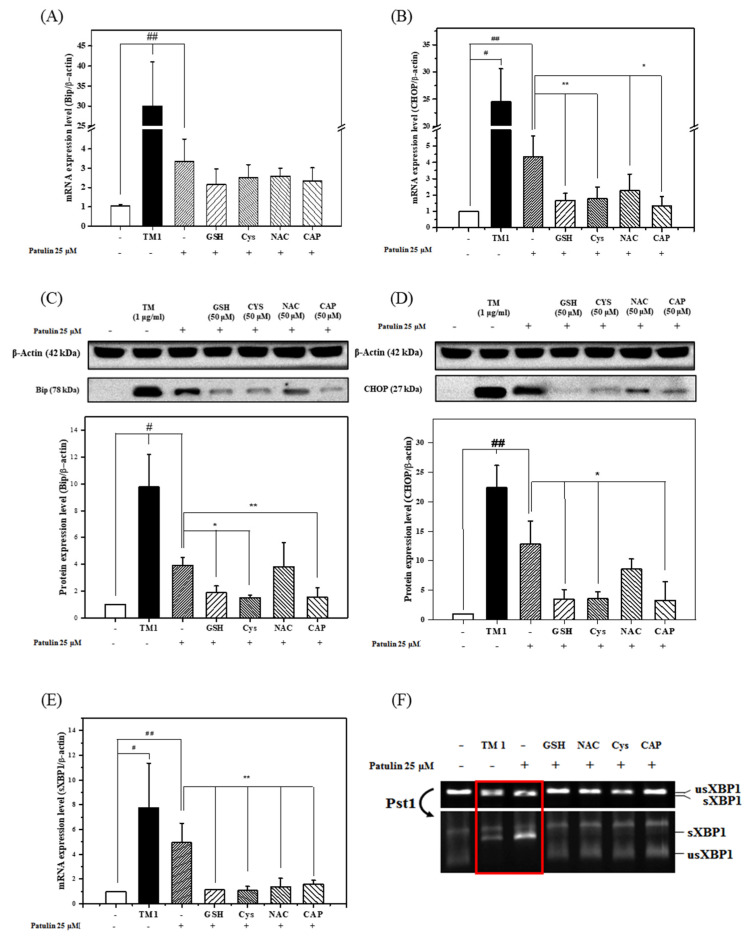
Expression of BiP, CHOP, and XBP1 and their protein levels in HepG2 cells after co-treatment with patulin (25 μM) and thiol-containing compounds (50 μM) for 48 h. GSH, glutathione; NAC, l-acetyl-l-cysteine; Cys, cysteine; CAP, captopril. The BiP (**A**) and CHOP (**B**) mRNA levels after patulin treatment were analyzed by reverse transcription quantitative real-time polymerase chain reaction. The BiP (**C**) and CHOP (**D**) protein levels after patulin treatment were analyzed by western blotting. (**E**) The effects of co-treatment with patulin (25 μM) and thiol-containing compounds on the expression of spliced XBP1 cut by PstI (**F**). The mRNA for β-actin (*ACTB*) was used as the housekeeping reference gene to normalize the data using the ΔΔCt method. The data are shown as the mean ± SEM from three independent experiments and were compared with the patulin-treated cells or the vehicle controls (0.5% DMSO). * *p* < 0.05 and ** *p* < 0.01 (*t*-test) compared with the patulin treatment (25 μM). # *p* < 0.05 and ## *p* < 0.01 (*t*-test) compared with the vehicle control (0.5% DMSO).

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
