# Peer review of "Natural Thiols, but Not Thioethers, Attenuate Patulin-Induced Endoplasmic Reticulum Stress in HepG2 Cells"

_toxins, 2021, doi:10.3390/toxins13100727_

Round 1

Reviewer 1 Report

Major corrections:

In Figure 2, given the fact that MTT is more metabolic assay, I would advise to use another cell toxicity assay (PI, celltoxgreen,…), at least to confirm the selected patulin concentrations

 In figure 3, although the results are convincing, a discrepancy exists between the western blot data and  mRNA expression results. The authors show a significant increase of ER stress markers at the mRNA level and not on the protein level even though they present an obvious difference in the signal intensity between the different studied conditions. Could the authors show whether the comparisons between the levels of ER stress markers are significant?

Also, I would advise normalizing spliced form of XBP1 to the unspliced one, rather than to actin (3E).

The panel 5F is not labelled.

Western blotting of β-actin seem often overexposed

Overall, the paper is focused only on the HepG2 model. I would recommend to confirm some results in another human liver cell line for reinforcing the scientific message and then change the title for hepatic cells.

Minor text corrections:

Line 159 - mRNA level measured with PCR or gene expression not gene level measured with PCR

Lines 174-175: given the fact that the ER stress markers didn’t decrease to the level of control, I would advise concluding that thiols decreased or partially inhibited patulin-induced ER stress.

In the discussion,  line 198 - part of this sentence is not supported with data

Author Response

Responses to the reviewer #1’s comments:

Major corrections:

Comment 1. In Figure 2, given the fact that MTT is more metabolic assay, I would advise to use another cell toxicity assay (PI, celltoxgreen,…), at least to confirm the selected patulin concentrations

esponse: Thank you for your careful review and comments. Regarding MTT assay, we agree with the reviewer's comment. As you know, this assay measures cell viability in terms of reducing activity, in which tetrazolium compounds are enzymatically converted to water-insoluble formazans by dehydrogenases occurring in the mitochondria of living cells. This assay is reported to have limitations, including interactions with other reducing chemicals, glucose concentrations in the cell culture medium, and involvement with enzymes located in other organelles such as ER. Nevertheless, we used this assay because it is still considered by many researchers to be the 'gold standard' for determining cell viability and proliferation. Furthermore, we used the MTT assay in this study to confirm and compare the IC50 of patulin in HepG2 cells by using the MTT assay in previous reports (Ayed-Boussema et al., 2011 [29] and Diao et al., 2018 [30]). Even though your suggestion is very good, another cell toxicity assay cannot be included in this manuscript due to the constraints of time, manpower, and research budget. However, it would be better to express the MTT assay more accurately, so in the revised manuscript, "cell viability" was changed into "cell metabolic activity" (line 99, 102, 104, 108, 152, 154-155, 157, 164, 166, 208, 214, and 334). We also added two sentences describing the limitations of this assay in the discussion section to give readers more information (line 216-223 in page 8). The added sentences read “In this study, cell viability was measured using MTT assay, which measures the metabolic activity of enzymatic reduction of tetrazolium compounds to water-insoluble formazans by dehydrogenases occurring in the mitochondria of living HepG2 cells. Although this assay is still used by many researchers to determine cell viability and proliferation, it is reported to have limitations, including interactions with other reducing chemicals, glucose concentrations in the cell culture medium, and involvement with enzymes located in other organelles such as ER [31]”. Accordingly, 1 reference (van Tonder et al., 2015) was added.

Comment 2. In figure 3, although the results are convincing, a discrepancy exists between the western blot data and mRNA expression results. The authors show a significant increase of ER stress markers at the mRNA level and not on the protein level even though they present an obvious difference in the signal intensity between the different studied conditions. Could the authors show whether the comparisons between the levels of ER stress markers are significant?

Response: Thank you for your comments. As the reviewers pointed out, we reviewed and performed statistical analyses of all mRNA and protein-related data of BiP and CHOP, which are key markers of ER stress. As a result, there was a good agreement between the western blot data and mRNA expression results for CHOP and statistical significance was found when compared to the control (p<0.05). BiP results also showed somewhat similar trends between the western blot data and mRNA expression results. However, quantification of Western blot results performed in 3 independent replicates using Land 1D software (version 4.0) was not statistically significant when compared to the control (p>0.05). This is considered an independent data error of cell-based bioassays. Nevertheless, the trends of three independent western blots appears to be recognizable, so we presented three independent western blots for BiP marker in a word file attached below for reviewer reference (Figure showing three independent western blots). In the revised manuscript, we presented a new Figure 3 indicating the statistical significance (Figure 3C and D in page 5).

Comment 3. Also, I would advise normalizing spliced form of XBP1 to the unspliced one, rather than to actin (3E)

Response: Thank you for your comments. In general, there are three methods to evaluate the increase of sXBP1. The first method is semi-qPCR, using an XBP1-specific primer capable of detecting both sXBP1 (263 bp) and uXBP1 (289 bp) by normalizing the ratio of the mRNA levels of sXBP1 to uXBP1 with a housekeeping gene such as actin. The second method uses PCR and the restriction endonuclease, Pst I. XBP1 mRNA contains a 26 bp intron, in turn containing the Pst I restriction enzyme, whereas sXBP1 mRNA does not contain this Pst I site because the intron has been removed. Thus, Pst I digestion cleaves uXBP1 into two small fragments, while sXBP1 migrates as one fragment without being cleaved by Pst I. This method allows for better discrimination between sXBP1 and uXBP1. The third method involves PCR performed using specific primers of uXBP1 and sXBP1, followed by loading the PCR products together. Although the third method is easier to identify sXBP1 because it is larger than 26 bp, there is a limitation that interpretation of the results can differ depending on the PCR conditions. As you suggested, for the first method, it would be better to detect both sXBP1 (263 bp) and uXBP1 (289 bp) by normalizing the ratio of the mRNA levels of sXBP1 to uXBP1 with a housekeeping gene such as actin. So, instead of using one method, we detected an increase in sXBP1 using two methods mentioned above. Because each method to evaluate the increase in sXBP1 has its limitation, in this study, two detection methods were used: a method of normalizing the mRNA level of sXBP1 with a housekeeping gene such as actin, and a method of cleavage with the restriction endonuclease Pst I.

Comment 4. The panel 5F is not labelled

Response: We apologize for our mistake. In response to reviewer’s indication, we labelled the panel 5F in the revised manuscript (Figure 5 in page 5).

Comment 5. Western blotting of β-actin seems often overexposed

Response: It also seems to us that β-actin has been overexposed. Fortunately, there was a figure that was less overexposed, so we replaced it (5C-D) (Figure 5 in page 7).

Comment 6. Overall, the paper is focused only on the HepG2 model. I would recommend to confirm some results in another human liver cell line for reinforcing the scientific message and then change the title for hepatic cells

Response: Thanks for the constructive suggestions. We are currently unable to conduct further experiments due to the constraints of time, manpower, and research budget. In the near future, if the circumstances permit, we plan to conduct further studies on the mechanism of action while confirming that the present results can be reproduced in various hepatic cells. So, instead of using the term “hepatic cells” in the title of the present study, we used specific term “HepG2 cells”.

Minor text corrections:
Comment 1. Line 159 - mRNA level measured with PCR or gene expression not gene level measured with PCR

Response: Thank you for your comment. In response to reviewer’s indication, we changed from “gene level” to “mRNA level” throughout the revised manuscript (line 113, 117, 119, 121, 122, 124, 143, 174, 186, 196, 200, 212, 264, 274, and 361).

Comment 2. Lines 174-175: given the fact that the ER stress markers didn’t decrease to the level of control, I would advise concluding that thiols decreased or partially inhibited patulin-induced ER stress

Response: In response to reviewer’s indication, we changed a sentence from “From these findings, we concluded that thiol-containing compounds inhibited patulin-induced ER stress in HepG2 cells” to “From these findings, we concluded that thiol-containing compounds decreased or partially inhibited patulin-induced ER stress in HepG2 cells” in the revised manuscript (line 189-191 in page 7).

Comment 3. In the discussion, line 198 - part of this sentence is not supported with data

Response: Thank you for your thoughtful review and comments. In response to reviewer’s indication, we modified a sentence in the revised manuscript (line 209-211 in page 8). The modified sentence reads “In addition, among the seven tested sulfur-containing compounds, thiols (GSH, NAC, Cys, and CAP) markedly reduced patulin-induced cytotoxicity, whereas thioethers (SulA, SulE and SAC) had no effect”. And, we deleted a sentence which was not supported with data. The deleted sentence reads “However, thioethers (SulA, SulE, and SAC) did not affect patulin-induced cytotoxicity and ER stress” (page 8).

Reviewer 2 Report

The study demonstrates the effect of patulin and thioethers on the markers of ER stress induction. The study is carefully and nicely executed, however there are several major points which deserve to be resolved before publication.

Please provide statistical analysis for Figure 2 [mtt results]

The narrative of the manuscript needs to me modified. MTT assay is dependent on activity of specific enzymes and therefore should be considered not as a viability test but as mitochondrial activity test

The authors pointed in the introduction that ER stress induction can be stimulated by three different branches; however, during study they focus mainly on one of them. Huge amount of literature data strongly suggests that all three branches strictly cooperate by several loops. Therefore, I suggest that the authors should supplement the results with data on possible role of other branches.

Please provide bands for housekeeping gene reflecting the experiment of XBP-1 splicing

Author Response

Responses to the reviewer #2’s comments: 

Comment 1. “The study demonstrates the effect of patulin and thioethers on the markers of ER stress induction. The study is carefully and nicely executed, however there are several major points which deserve to be resolved before publication.

Please provide statistical analysis for Figure 2 [mtt results]”

Response: Thank you for your careful review and comments. In response to reviewer’s indication, we provided statistical analysis for Figure 2 in the revised manuscript (Figure 2 in page 3). Accordingly, “**p<0.01 (t-test) compared with the vesicle control group” was added in the caption of Figure 3.

Comment 2. The narrative of the manuscript needs to me modified. MTT assay is dependent on activity of specific enzymes and therefore should be considered not as a viability test but as mitochondrial activity test

Response: Thank you for your careful review and comments. Regarding MTT assay, we agree with the reviewer's comment. As you know, this assay measures cell viability in terms of reducing activity, in which tetrazolium compounds are enzymatically converted to water-insoluble formazans by dehydrogenases occurring in the mitochondria of living cells. However, this assay is reported to have limitations, including interactions with other reducing chemicals, glucose concentrations in the cell culture medium, and involvement with enzymes located in other organelles such as ER. Nevertheless, we used this assay because it is still considered by many researchers to be the 'gold standard' for determining cell viability and proliferation. Because we agree with the reviewers that the MTT assay should be accurately expressed, we changed from "cell viability" to "cell metabolic activity" in the revised manuscript (line 99, 102, 104, 108, 152, 154-155, 157, 164, 166, 208, 214, and 334). Furthermore, we also added two sentences describing the limitations of this assay in the discussion section to give readers more information (line 216-223 in page 8). The added sentences read “In this study, cell viability was measured using MTT assay, which measures the metabolic activity of enzymatic reduction of tetrazolium compounds to water-insoluble formazans by dehydrogenases occurring in the mitochondria of living HepG2 cells. Although this assay is still used by many researchers to determine cell viability and proliferation, it is reported to have limitations, including interactions with other reducing chemicals, glucose concentrations in the cell culture medium, and involvement with enzymes located in other organelles such as ER [31]”. Accordingly, 1 reference (van Tonder et al., 2015) was added.

Comment 3. The authors pointed in the introduction that ER stress induction can be stimulated by three different branches; however, during study they focus mainly on one of them. Huge amount of literature data strongly suggests that all three branches strictly cooperate by several loops. Therefore, I suggest that the authors should supplement the results with data on possible role of other branches

Response: Thanks for your constructive suggestions. As suggested by the reviewers, we also want to supplement our results with data on the possible roles of the three branches (PERK, ATF6 and IRE1a-pathway). However, the aims of this study are to investigate (1) whether ER stress is induced in HepG2 cells, an in vitro liver toxicity model, in response to patulin, and (2) if so, whether it can be alleviated by seven naturally occurring sulfur-containing compounds (four thiols (R-SH) namely glutathione, cysteine, captopril , and N-acetyl-L-cysteine, and three thioethers (R-S-R¢) namely S-allyl-L-cysteine, sulforaphane A, and sulforaphane E. Previously, Boussabbeh et al. (2015) reported that patulin induces ER stress by increasing representative markers of ER stress such as BiP, CHOP, and spliced XBP1 in kidney cells. Therefore, we wanted to check whether the same trend was observed in HepG2, which was used as a liver cell model. Next, we tried to figure out whether the seven naturally occurring sulfur-containing compounds could alleviate the ER stress. Therefore, we plan to conduct a follow-up study to determine which pathways or interactions produce these effects. We are currently unable to conduct further experiments due to the constraints of time, manpower, and research budget. In the near future, if circumstances permit, we will conduct further studies on confirming that the present results can be reproduced in various hepatic cells as well, and the mechanism of action, including possible roles of the three branches (PERK, ATF6, IRE1a-pathway). In order to clarify this, we changed a sentence from “In addition, the precise mechanism whereby thiol compounds attenuate patulin-induced ER stress should be elucidated” to “In addition, the possible roles of the three branches (PERK, ATF6, IRE1a-pathway) in patulin-induced ER stress and precise mechanism of thiol compounds should be elucidated” in the revised manuscript (line 305-307 in page 9).

Comment 4. Please provide bands for housekeeping gene reflecting the experiment of XBP-1 splicing

Response: Thank you for your suggestion. In general, there are three methods to evaluate the increase in sXBP1. The first method is semi-qPCR, using an XBP1-specific primer capable of detecting both sXBP1 (263 bp) and uXBP1 (289 bp) by normalizing the ratio of the mRNA levels of sXBP1 to uXBP1 with a housekeeping gene such as actin. The second method uses PCR and the restriction endonuclease, Pst I. XBP1 mRNA contains a 26 bp intron, in turn containing the Pst I restriction enzyme, whereas sXBP1 mRNA does not contain this Pst I site because the intron has been removed. Thus, Pst I digestion cleaves uXBP1 into two small fragments, while sXBP1 migrates as one fragment without being cleaved by Pst I. This method allows for better discrimination between sXBP1 and uXBP1. The third method involves PCR performed using specific primers of uXBP1 and sXBP1, followed by loading the PCR products together. Although the third method is easier to identify sXBP1 because it is larger than 26 bp, there is a limitation that interpretation of the results can differ depending on the PCR conditions. Because each method to evaluate the increase of sXBP1 has its limitation, in this study, two detection methods were used: a method of normalizing the mRNA level of sXBP1 with a housekeeping gene such as actin, and a method of cleavage with the restriction endonuclease Pst I. In Figures 3E and 5E, the mRNA level of sXBP1 was normalized with a housekeeping gene beta-actin. In Figures 3F and 5F, we confirmed the increase in sXBP1 in the same sample using a method of cleavage with the restriction endonuclease Pst I without detecting housekeeping gene.

In addition to the responses shown above, we also adjusted all references according to the Journal style (Penicillium expansumà Penicillium expansum in line 420; Escherichia colià Escherichia coli in line 446; delete “,” in line 474 etc.) and corrected a mistake (“BiP (78 kDa”à “CHOP (27 kDa)” in Figure 5D) in the revised manuscript.

I would like to thank for reviewers’ comments and suggestions again. We mark all the changes in red in the revised manuscript. We think that the quality of our manuscript has been highly improved by the process of review and revision. I hope we have addressed all points to your and reviewer’s satisfaction. I am looking forward to hearing from you soon.

Round 2

Reviewer 1 Report

Dear authors,

I would like to thank the authors for their reply. Most of my concerns have been considered.

However, I am still doubtful about the relevance  of data obtained in only one model. I definitely suggest to confirm few results in another model.

Author Response

(Manuscript #Toxins-1355390,  )

All authors greatly appreciate the comments given by the reviewer. We thoroughly took into consideration the reviewer’s comment. Followings are our responses to the comment.

Response to the reviewer #1’s comment:

Comment 1. I would like to thank the authors for their reply. Most of my concerns have been considered. However, I am still doubtful about the relevance of data obtained in only one model. I definitely suggest to confirm few results in another model.

Response: Thanks for your comment. We generally agree with your comment that results observed in one cell model should be confirmed in another cell model. However, we are obliged to your comment as follows in terms of 1) the purpose of this study, 2) the hepatic cell model, and 3) the time, manpower, and budget required for additional experiments.

1) Purpose of the study: As you know, the aims of this study are to investigate (1) whether ER stress is induced in HepG2 cells, an in vitro liver toxicity model, in response to patulin, and (2) if so, whether it can be alleviated by seven naturally occurring sulfur-containing compounds (four thiols (R-SH) namely glutathione, cysteine, captopril , and N-acetyl-L-cysteine, and three thioethers (R-S-R') namely S-allyl-L-cysteine, sulforaphane A, and sulforaphane E. Previously, Boussabbeh et al. (2015) reported that patulin induces ER stress by increasing representative markers of ER stress such as BiP, CHOP, and spliced XBP1 in kidney cells. Based on this, we hypothesized that a similar trend would be observed in hepatic cells where toxic substances such as patulin were metabolized. As a first-stage study, we wanted to confirm whether the same trend was observed using HepG2 as a hepatic cell model. Next, we tried to figure out whether the seven naturally occurring sulfur-containing compounds could alleviate the ER stress.

Therefore, our follow-up study will be on the mechanism of action of the observed results, confirming whether the observed results in the HepG2 cell model used in the first step are the same in other hepatocytes (possibly normal human hepatocytes). In the first revised manuscript (line 302-307 in page 9), we added sentences describing these follow-up studies. These sentences read “However, because the results of this study were obtained using an in vitro cell model, these effects should further be confirmed in other in vitro hepatic cell models, in vivo animal models or human studies. In addition, the possible roles of the three branches (PERK, ATF6, and IRE1α pathways) in patulin-induced ER stress and precise mechanism of thiol compounds should be elucidated”.

2) The hepatic cell model:

We believe the reviewer's comment was given from the concern that the results presented in this study may be specific to HepG2 cell model and may not be observed in another hepatic cell model. At the beginning of this study, we performed MTT assay using Chang cells (recognized as a hepatic cell line that is closer to normal cells) and HepG2 cells (recognized as a cell line commonly used in hepatotoxicity studies) to select a cell model. In addition, mRNA levels of BiP and CHOP, which are representative markers of ER stress, were evaluated by qRT-PCR. As a result, mitochondrial metabolic activity decreased as patulin concentration increased in both cell models. Also, mRNA levels of BiP and CHOP increased as the patulin concentration increased. However, a decrease in cellular metabolic activity and an increase in mRNA levels of BiP and CHOP were observed at higher patulin concentrations in Chang cells in comparison with HepG2. However, we found that the Chang cell line was HeLa-contaminant based on isoenzyme analysis, HeLa marker chromosomes, and DNA fingerprinting even though it was originally known as the normal liver-derived cell line (Gao et al., Hepatology). Thus, we selected HepG2 cells as a hepatic cell model for our study. Also, the results obtained from the misidentified Chang cells could no longer be presented. Nevertheless, these results suggest that the induction of ER stress by patulin may be not specific to the HepG2 cells used as a hepatic cell model, but also may appear in Chang cells, which have, at least in part, the characteristics of hepatic cells. Therefore, we plan to confirm the results of this study and carry out the mechanism of action using a normal hepatic cell model as a follow-up study. For the reviewer’s reference, preliminary results obtained from the Chang cell model performed at the beginning of the study were presented as a file below.

3) The time, manpower, and budget required for experiments using another model.

We think it is another form of study to confirm the results in other cell models commented by the reviewer. It is a time-consuming and expensive experiment to perform the MTT assay at least three times in a different cell model and to evaluate the markers related to ER stress. In addition, as the rationale of the hypothesis and purpose of this study were indicated above, we think that the hypothesis testing by using one cell model in the first stage of the study is generally accepted research as science-based research. We think that a study to determine whether the results obtained in one cell model are reproduced in another cell model may not be done necessarily in one paper (study). We have often seen this research scheme proceeds in this way in a variety of published reports.

We are currently unable to conduct further experiments using another cell model due to the constraints of time, manpower, and research budget. Soon, if the circumstances permit, we plan to conduct further studies on the mechanism of action, confirming that the present results can be reproduced in various hepatic cells. We are now applying for research grants for these follow-up studies. Again, first, we specified “HepG2 cells” instead of “hepatic cells” in the title because we studied using one model (line 2-3 in page 1). Second, in the discussion section of the manuscript, the characteristics and limitations of HepG2 cells were described (line 224-230 in page 8). Third, as a follow-up study, it was stated that a mechanism study would be required in addition to confirming whether the results obtained from this study might be observed in other hepatic cell models (line 302-307 in page 9).

I would like to thank for reviewer’s comments and suggestions again. We think that the quality of our manuscript has been highly improved by the two rounds of process of review and revision. For the comment to confirm the results obtained in this study using other cell model, please consider the time and cost of additional experiments, and the limitation that the revision of this manuscript should be done within 4 days. I am looking forward to hearing from you soon.

Sincerely Yours,

Authors

Reviewer 2 Report

the authors have improved the manuscript, therefore it deserves to be published

Author Response

We would like to thank for reviewer’s comments and suggestions again. We think that the quality of our manuscript has been highly improved by your comments and suggestions. 

Round 3

Reviewer 1 Report

The authors replied to my concerns.